# Risk factors for cerebral complications in patients with pulmonary arteriovenous malformations: A multicenter retrospective cohort study

Young-Ok Na[1,2], Hwa Kyung Park[1,2], Jae-Kyeong Lee[1,2], Bo-Gun Kho[1,2], Tae-Ok Kim[1,2], Hong-Joon Shin[1,2]*, Yong-Soo Kwon[1,2], Yu-Il Kim[1,2], Sung-Chul Lim[1,2], Hyung-Joo Oh[1,2], Cheol-Kyu Park[2,3], In-Jae Oh[2,3], Young-Chul Kim[2,3], Ha-Young Park[2,4]

1 Department of Internal Medicine, Chonnam National University Hospital, Gwangju, Republic of Korea, 2 Chonnam National University Medical School, Gwangju, Republic of Korea, 3 Lung and Esophageal Cancer Clinic, Chonnam National University Hwasun Hospital, Joennam, Republic of Korea, 4 Department of Internal Medicine, Chonnam National University Bitgoeul Hospital, Gwangju, Republic of Korea

* 99naussica@naver.com

**Data Availability Statement:** All relevant data are within the manuscript and its Supporting Information files.

## Abstract

### Objective

Pulmonary arteriovenous malformation (PAVM) is a rare pulmonary disease. Although most patients with PAVMs are asymptomatic, cerebral complications associated with PAVMs are often fatal. This study aimed to evaluate the risk factors for cerebral complications in patients with PAVMs.

### Methods

We retrospectively reviewed the medical charts of patients with PAVMs between 2003 and 2021 at two tertiary referral hospitals and one secondary hospital.

### Results

Fifty-five patients diagnosed with PAVMs were enrolled in this study. Most patients were female (89.1%), and the median age was 53 years. Thirty patients (54.5%) had incidentally detected PAVMs without symptoms. Twenty-four patients (43.7%) with PAVMs were treated with embolotherapy or surgery. Thirteen patients (23.6%) had cerebral complications. There was no significant difference in the development of cerebral complications according to treatment; however, older age ($\geq$ 65 years) was associated with the development of new cerebral complications in untreated patients with PAVMs (odds ratio, 17.09; 95% confidence interval, 1.16–250.31; P = 0.038).

### Conclusion

Older age ($\geq$ 65 years) was a risk factor for the development of cerebral complications in patients with PAVMs; therefore, treatment should be considered in older patients with PAVMs.

**Funding:** This study was supported by the National Research Foundation of Korea funded by the Korean Government (grant 2019R1F1A1060899). The funders had no role in study design, data collection and analysis, decision to publish, or preparation of the manuscript.

**Competing interests:** The authors have declared that no competing interests exist.

## Introduction

Pulmonary arteriovenous malformation (PAVM) is a rare pulmonary disease, characterized by abnormal direct vascular communications between pulmonary arteries and veins resulting in right-to-left shunts [1]. The most common cause of PAVMs is hereditary hemorrhagic telangiectasia (HHT), although it varies by region and race [1–5].Most patients with PAVMs are asymptomatic; however, symptoms such as dyspnea, hemoptysis, chest discomfort, and neurological symptoms may accompany PAVMs [4, 6–10].

Serious complications of PAVMs include hypoxia, hemothorax, stroke, and brain abscess [6–8, 10, 11]. Treatments for PAVM, such as embolotherapy, reduce the development of PAVM complications [12]. Cerebral complications are reported in 9% to 41% of patients with PAVM, which can be fatal [6–11]. The risk of cerebral complications is low in younger patients, but high in patients with multiple PAVMs [13, 14]. The risk factor of cerebral complications associated with PAVM is not well known.

This study aimed to evaluate the risk factors associated with cerebral complications in patients with PAVMs.

## Patients and methods

### Study design and population

We retrospectively reviewed the medical charts of patients with PAVMs between January 2003 and May 2021 at two tertiary referral hospitals and one secondary hospital. We screened patients with PAVMs using diagnostic code, chest computed tomography (CT) findings, or pulmonary angiography findings during the study period. PAVMs were diagnosed by chest CT with enhancement or pulmonary angiography.

### Data collection

We investigated the patients' age, sex, underlying diseases (hypertension and diabetes), mode of PAVM detection (initial symptoms or incidentally), and treatments. We also investigated the types and feeding artery diameters of PAVMs and presence of HHT. The diagnoses of HHT were made according to the Curacao diagnostic criteria for HHT [15] as follows: spontaneous, recurrent nose bleeds; multiple telangiectasis, especially in the lips, oral cavity, fingers, and nose; visceral lesions such as gastrointestinal telangiectasia; hepatic and cerebral arteriovenous malformations; and a first-degree relative with HHT. Definite HHT was classified as patients who fulfilled three or more of the above-mentioned criteria. Probable HHT was classified as patients with two criteria fulfillments. The feeding artery and venous sac diameters were measured [16]. A modified Rankin Scale (mRS) score was used to evaluate the degree of disability or dependence in daily activities in patients with PAVM with cerebral complications [17]. The cerebral complications associated with PAVM were divided into cerebral ischemia, hemorrhage, and abscess. To determine whether there had been any brain complications before PAVM diagnosis, we reviewed the patient's medical records. A brain CT or MRI was used to diagnose cerebral complications associated with PAVM at the time of initial PAVM diagnosis or after PAVM was detected.

### Types of PAVMs

We classified the PAVMs as single and multiple types. Multiple PAVMs were defined as at least two PAVMs observed on imaging findings.

### Definition

Patients with incidentally detected PAVM were those whose PAVM was detected during a health screening, preoperative evaluation, or evaluation of other diseases, and presented asymptomatically.

### Evaluation of recanalization after embolotherapy

Patients who had a follow-up chest CT scan after embolotherapy were analyzed to evaluate recanalization. An evaluation of recanalization was performed on patients who had a follow-up chest CT scan after embolotherapy [6, 7, 18]. Recanalization was determined by 70% criteria following embolotherapy, which means less than 70% regression of the PAVM sac and draining vein [18].

### Ethics statement

The authors are accountable for all aspects of the work in ensuring that questions related to the accuracy or integrity of any part of the work are appropriately investigated and resolved. The protocol which was conducted according to the principle expressed in the Declaration of Helsinki (as revised in 2013). The Institutional Review Board at Chonnam National University Hospital (Gwangju, Republic of Korea) approved the study protocol and permitted the review and publication of our findings, as well as that of information derived from patient records (CNUH 2022–055) requirement for informed consent was waived because of the retrospective nature of the study, and approved by the Ethics Committee. Patient information was fully rendered innominate before the analysis.

### Statistical analyses

All data were expressed as medians with interquartile ranges or numbers (percentages). Factors associated with cerebral complications were selected by univariate logistic regression analysis. Subsequent multivariate logistic regression analyses included variables with P values $< 0.2$ in the univariate analysis using a backward method. We used Kaplan–Meier analysis to evaluate the time without development of cerebral complications. Factors associated with cerebral complications were identified using Cox-regression analysis that included variables with $P < 0.2$ in the univariate analysis using the backward method. All statistical analyses were performed using SPSS version 25.0 (IBM, Armonk, NY, USA); a P value of $<0.05$ was considered statistically significant.

## Results

A total of 70 individuals were screened, of which 15 were excluded for the following reasons: a CT or angiography did not reveal PAVM (n = 11), or they were diagnosed with pulmonary sequestration (n = 3) or a pulmonary venous anomaly (n = 1). Therefore, a total of 55 patients with PAVM were enrolled in this study.

### Patient characteristics

Fifty-five patients diagnosed with PAVMs were enrolled in this study. The median follow-up duration was 20.4 months (interquartile range, 3.3–50.5 months). The baseline characteristics of the study patients are shown in Table 1. Most patients were female (89.1%), and the median age was 53 years. The most common type of PAVMs was the single type (76.4%). Only one patient had HHT. Thirty patients had incidentally detected PAVM via the following: health screening (n = 18), preoperative evaluation (n = 6), or evaluation of other diseases (n = 6). A

**Table 1. Baseline characteristics of patients with pulmonary arteriovenous malformations.**

| Variables | n (%) |
|---|---|
| Total | |
| Male | 6 (10.9) |
| Female | 49 (89.1) |
| Age, yrs | |
| Median | 53 |
| IQR | 45–62 |
| Types of PAVMs | |
| Single | 42 (76.4) |
| Multiple | 13 (23.6) |
| Location of PAVMs | |
| Right upper lobe | 11 (20.0) |
| Right middle lobe | 10 (18.2) |
| Right lower lobe | 14 (25.5) |
| Left upper lobe | 19 (34.5) |
| Left lower lobe | 15 (27.3) |
| Hereditary hemorrhagic telangiectasia | 1 (1.8) |
| Feeding artery diameter, mm | |
| Median | 4.1 |
| IQR | 3.3–4.8 |
| Venous sac diameter, mm | |
| Median | 11.8 |
| IQR | 8.4–17.3 |
| Underlying disease | |
| Diabetes | 3 (5.5) |
| Hypertension | 11 (20.0) |
| Causes of detection | |
| Incidentally | 30 (54.5) |
| Coughing | 7 (12.7) |
| Neurologic symptoms | 4 (7.3) |
| Chest pain | 4 (7.3) |
| Hemoptysis | 3 (5.5) |
| Dyspnea | 3 (5.5) |
| Pneumonia | 2 (3.6) |
| Others | 2 (3.6) |
| Cerebral complications | 13 (23.6) |
| Cerebral ischemia | 11 (84.6) |
| Cerebral hemorrhage | 1 (7.7) |
| Cerebral abscess | 1 (7.7) |
| Treatment | |
| Observation | 31 (56.4) |
| Transcatheter embolization | 21 (38.2) |
| Surgery | 3 (5.5) |

IQR, interquartile range; PAVM, pulmonary arteriovenous malformation

total of thirteen patients (23.6%) had cerebral complications, including cerebral ischemia (n = 11), cerebral hemorrhage (n = 1), and cerebral abscess (n = 1). Of the 13 patients with cerebral complications, six developed these complications before their PAVM diagnosis, five developed these complications concurrently with their PAVM diagnosis, and two developed these complications after their PAVM diagnosis. Twenty-one (38.2%) patients underwent transcatheter embolization for PAVMs. Of these, two patients were not treated at the time of PAVM diagnosis and underwent embolotherapy after developing cerebral complications. Three patients (5.5%) underwent surgery for PAVMs. Cerebral complications occurred in four patients who did not undergo treatment after a PAVM diagnosis, and two patients had previous cerebral complications. Cerebral complications in all four patients included cerebral ischemia, and one patient showed severe functional impairments, with an mRS score of 5. The other three patients, however, only showed mild impairments with mRS scores of 1–3 (S1 Table). None of the patients who underwent treatment for PAVM developed further cerebral complications during the follow-up period.

A follow-up chest CT scan was performed on 13 of the 21 PAVM patients who underwent embolotherapy, and among them, recanalization was observed in two.

Fig 1 shows the number of patients diagnosed with PAVM in the three hospitals at 5-year intervals since 2003. Each year, the number of newly diagnosed PAVM patients has increased. In particular, the number of patients diagnosed incidentally without symptoms associated with PAVM has increased.

## Risk factors for cerebral complications associated with PAVMs

We evaluated the risk factors for cerebral complications, including cerebral hemorrhage and ischemia, at the time of PAVM diagnosis (Table 2). Multivariate logistic regression analysis showed older age ($\geq$ 65 years) was associated with increased cerebral complications (odds ratio [OR], 23.18; 95% confidence interval [CI], 2.61–205.45; P = 0.005)(Table 2). In contrast, patients with incidentally detected PAVMs showed decreased rates of cerebral complications (OR 0.14; 95% CI, 0.02–0.88; P = 0.037). Furthermore, the incidence of cerebral complications was low in incidentally detected patients.

Fig 2 shows the Kaplan–Meier curve for the development of new cerebral complications in the treated and untreated groups after PAVM diagnosis.

The untreated group tended to develop more cerebral complications during the follow-up period; however, there was no significant difference between the two groups (log-rank P = 0.088).

## Subgroup analysis of risk factors for new cerebral complications among untreated patients with PAVMs

We investigated the risk factors associated with new cerebral complications in 33 patients who did not receive treatment after their PAVM diagnosis. Multivariate logistic regression analysis showed older age ($\geq$65 years) was associated with increased cerebral complications (OR, 17.09; 95% CI, 1.16–250.31; P = 0.038) (Table 3).

Fig 3 shows the Kaplan–Meier curve for the development of new cerebral complications in the $\geq$ 65 years and < 65 years age groups of patients who did not undergo treatment after a PAVM diagnosis. The older age group ($\geq$65 years) was significantly associated with the development of cerebral complications during the follow-up period (log-rank P = 0.014).Multivariate Cox-regression analysis showed that the older age group ($\geq$65 years) was associated with an increased risk of cerebral complications (hazard ratio, 18.00; 95% CI, 1.48–218.95; P = 0.023).

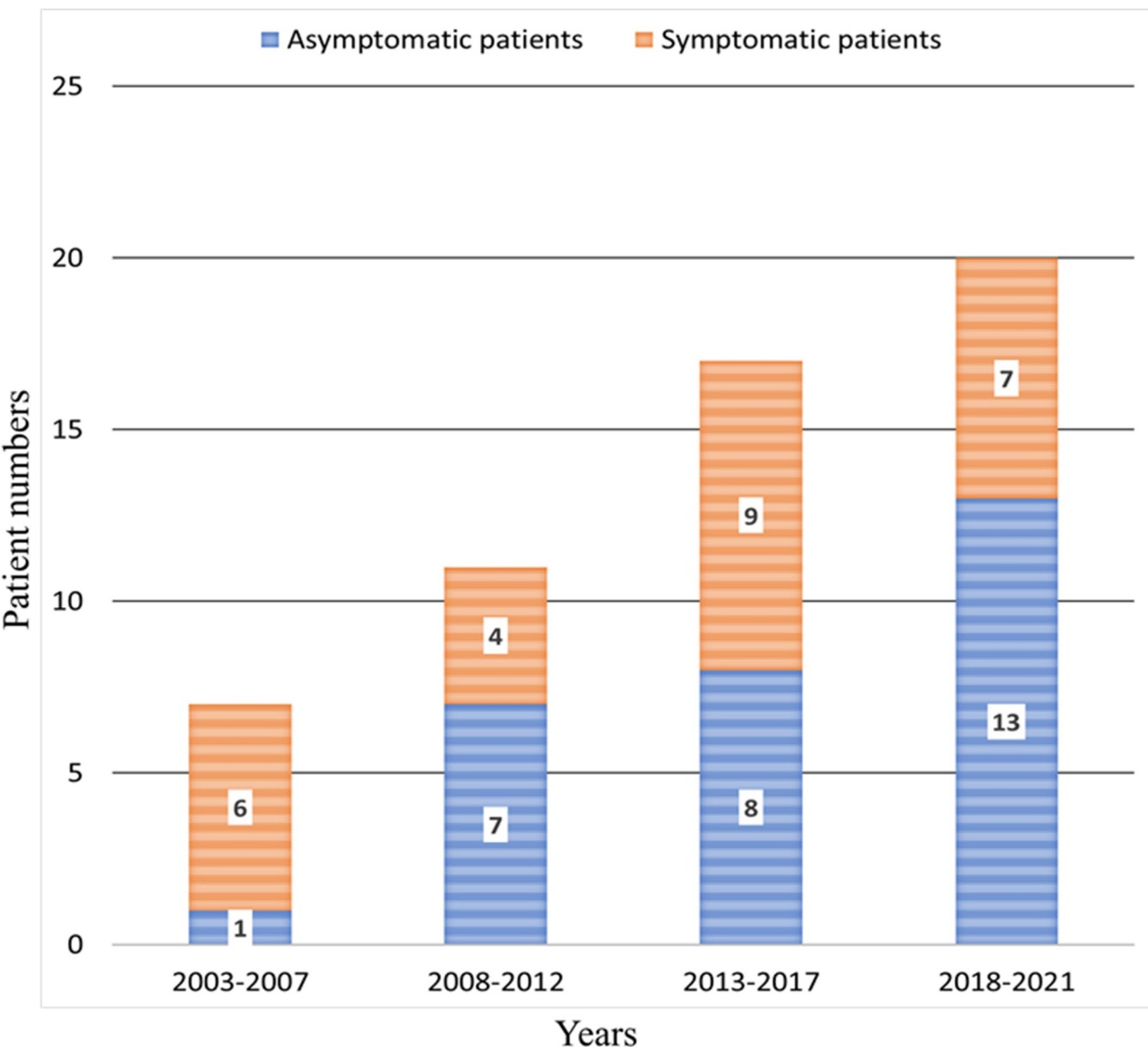

**Fig 1. Number of patients diagnosed with pulmonary arteriovenous malformation from 2003 to 2021.**

## Discussion

We described the risk factors for cerebral complications in patients with PAVMs in this multi-center retrospective cohort study. The number of newly diagnosed PAVM patients is increasing; in particular, the number of patients diagnosed incidentally without symptoms has increased. Older age ($\geq$ 65 years) was associated with increased cerebral complications. The untreated group tended to develop more cerebral complications during the follow-up period; however, there was no significant difference between the two groups. The older age group ($\geq$65 years) was significantly associated with the development of cerebral complications during the follow-up period among patients who did not undergo treatment for PAVMs.

**Table 2. Risk factors associated with cerebral complications in patients with pulmonary arteriovenous malformations (n = 55).**

| Variables | OR | 95% CI | P value |
|---|---|---|---|
| **Univariate analysis** | | | |
| Age ≥ 65 | 10.85 | 2.18–53.95 | 0.004 |
| Male | < 0.00 | 0.00–0.00 | 0.999 |
| Incidental detection | 0.25 | 0.06–0.97 | 0.046 |
| Hypertension | 3.75 | 0.91–15.40 | 0.067 |
| Diabetes | 7.45 | 0.61–90.00 | 0.114 |
| Feeding artery diameter | 1.08 | 0.76–1.53 | 0.647 |
| Venous sac diameter | 0.98 | 0.90–1.08 | 0.804 |
| Types of PAVM | 0.96 | 0.22–4.18 | 0.957 |
| Location of PAVM | | | |
| Right upper lobe | 0.26 | 0.31–2.31 | 0.230 |
| Right middle lobe | 0.77 | 0.14–4.19 | 0.765 |
| Right lower lobe | 1.42 | 0.35–5.62 | 0.616 |
| Left upper lobe | 0.80 | 0.21–3.04 | 0.743 |
| Left lower lobe | 2.00 | 0.53–7.51 | 0.305 |
| **Multivariate analysis** | | | |
| Age ≥ 65 | 23.18 | 2.61–205.45 | 0.005 |
| Incidental detection | 0.14 | 0.02–0.88 | 0.037 |
| Hypertension | 7.53 | 0.86–65.50 | 0.067 |
| Diabetes | <0.00 | 0.00–0.00 | 0.999 |

OR, odds ratio; CI, confidence interval; PAVM, pulmonary arteriovenous malformation

Most patients with PAVMs had respiratory symptoms according to Gossage et al. [3]. However, subsequent studies have reported that most patients with PAVMs are asymptomatic [4, 6–10]. In this study, 54.5% of patients with PAVMs were asymptomatic, and PAVMs were detected incidentally. In particular, the number of patients with incidentally detected PAVM had increased over time in the follow-up cohort. The increase in the number of patients with incidentally detected PAVM is most likely due to the incidental detection of PAVMs on chest imaging or during preoperative evaluations [6–10]. We found that the majority of patients (54%) were diagnosed with PAVM incidentally during similar health screenings. Transthoracic contrast echocardiography (TTCE) is recommended as a screening test for PAVMs in patients with a high risk of PAVMs or suspected right-to-left shunts [15, 19–21].The sensitivity and specificity of TTCE for detecting right-to-left shunts are high [19, 20, 22]. Compared to chest CT, TTCE does not expose the patient to radiation [15]. When a right-to-left shunt is confirmed with TTCE, a chest CT or angiography is performed to confirm the PAVM [15, 23]. Although Chest CT and TTCE examined together exhibit almost 100% sensitivity and negative predictive value [19, 20], it is unclear whether TTCE should be performed in patients withincidentally detected PAVMs on chest CT scans. In this study, only four patients (7.3%) underwent TTCE. Because additional TTCE may not be necessary in patients with PAVM detected incidentally on chest CT, further prospective studies are needed.

HHT is an autosomal dominant disease. Patients with PAVM account for 15% to 35% of all patients with HHT [15]. In North America and Europe, 60% to 80% of PAVM patients were found to have HHT [1–3]. In Japan, according to Shioya et al, patients with HHT and PAVMs accounted for 15% of PAVM patients [5]. Kim et al. reported that HHT-related PAVM occurred in 13% of cases in South Korea [4]. In the present study, there was only one patient

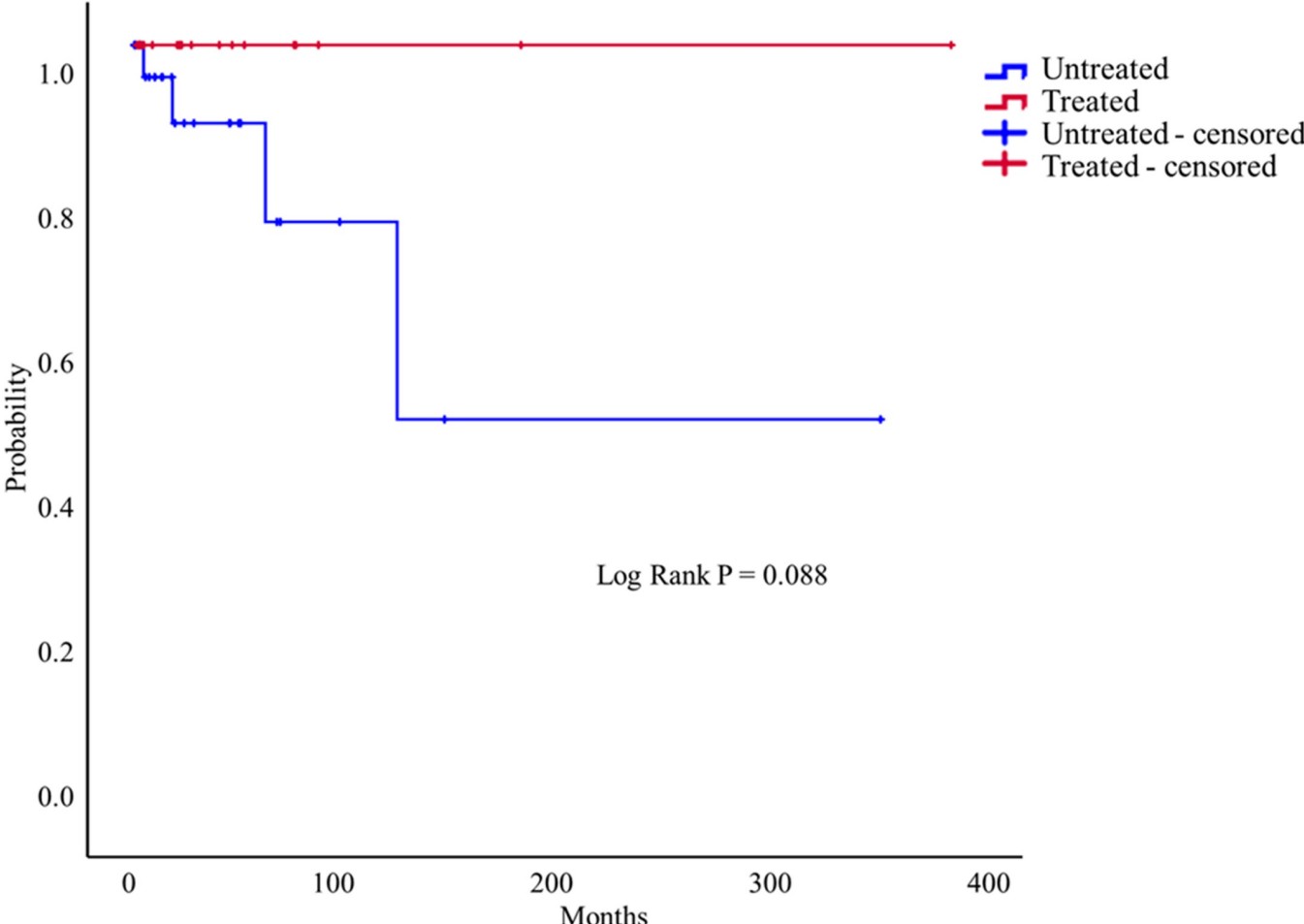

**Fig 2. Kaplan–Meier curve illustrating the time without the development of new cerebral complications associated with PAVM treatment.**

**Table 3. Risk factors for new cerebral complications in untreated patients after PAVMs were detected (n = 33).**

| Variables | OR | 95% CI | P value |
|---|---|---|---|
| Univariate analysis | | | |
| Age ≥ 65 | 18.00 | 1.48–218.95 | 0.023 |
| Male | < 0.00 | 0.00–0.00 | 0.999 |
| Incidental detection | 0.21 | 0.02–2.34 | 0.208 |
| Hypertension | 4.80 | 0.54–42.63 | 0.159 |
| Feeding artery diameter | 0.80 | 0.34–1.86 | 0.608 |
| Venous sac diameter | 1.01 | 0.81–1.24 | 0.947 |
| Types of PAVM | < 0.00 | 0.00–0.00 | 0.999 |
| Multivariate analysis | | | |
| Age ≥ 65 | 17.09 | 1.16–250.31 | 0.038 |
| HTN | 7.68 | 0.47–125.23 | 0.152 |

OR, odds ratio; CI, confidence interval; PAVM, pulmonary arteriovenous malformation

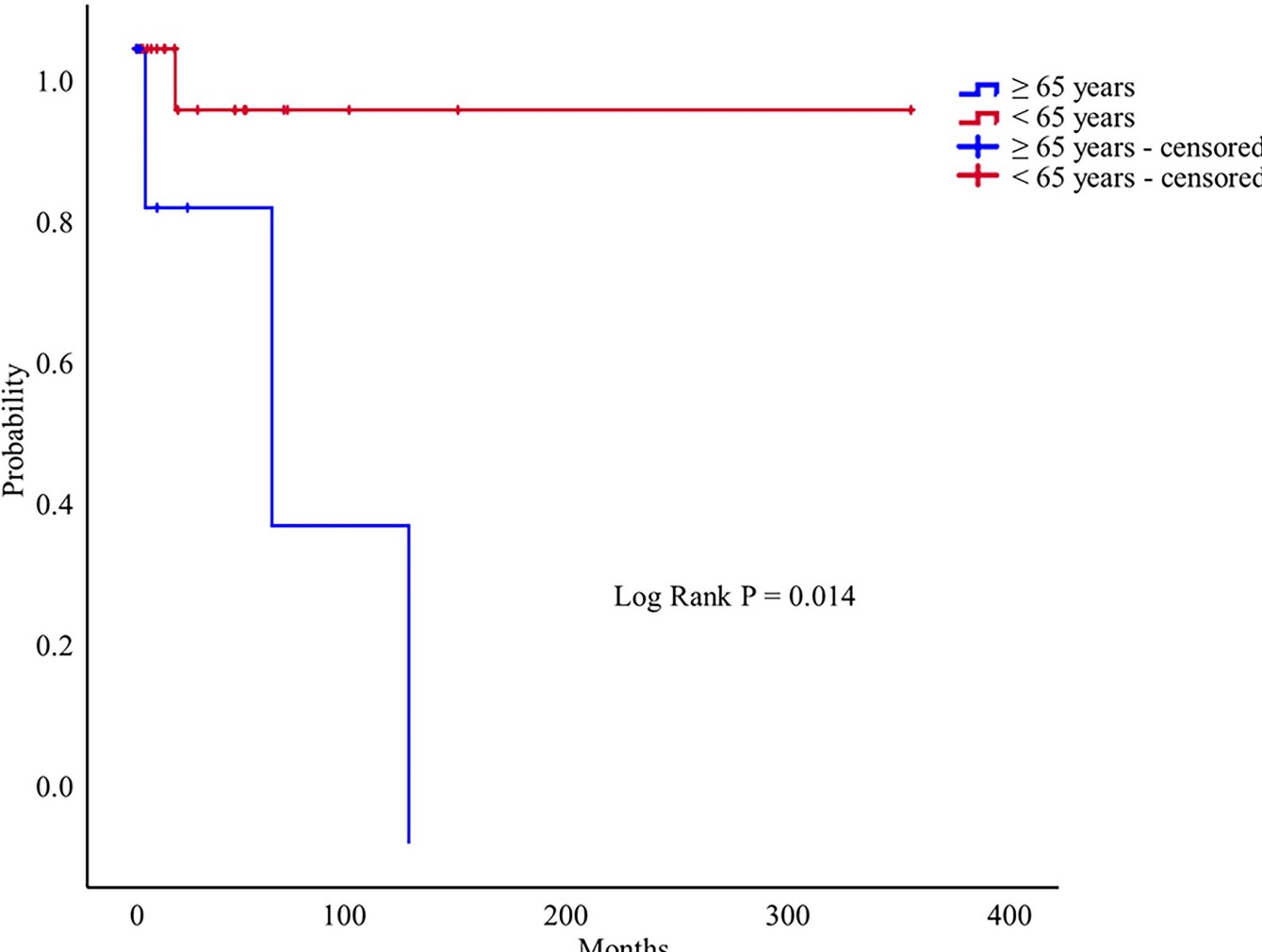

**Fig 3. Kaplan–Meier curve illustrating the time without the development of new cerebral complications in untreated PAVM patients (n = 33).**

with HHT. PAVM is rarely reported in Asia, which may be attributable to racial differences [15, 24]. In this retrospective study, HHT may have been underestimated in patients with PAVM. In order to avoid fatal complications, systematic education and training for the diagnosis and treatment of PAVMs in patients with HHT are needed [25].

Life-threatening complications of PAVMs include stroke, transient ischemic attack, cerebral abscess, and massive hemoptysis. Between 9% and 41% of patients with PAVMs develop cerebral complications, mainly stroke and cerebral abscess [6–11]. Patients with multiple PAVMs and older patients were at greater risk of cerebral complications [13, 14]. According to various studies, cerebral complications are more likely to occur in patients with PAVMs with feeding artery diameters greater than 3 mm [2, 13].Some previous reports suggest that cerebral complications are not directly related to feeding artery diametersize [9, 26]. Patients with feeding artery diameters of 2 mm are also recommended for treatment in the recent PAVM treatment guidelines [15]. There was no statistical significance between feeding artery diameter size and cerebral complications in this study. In patients over 65 years of age, the odds of developing cerebral complications were 16 times higher than that in patients under 65 years of age. In this study, incidentally detected PAVM patients had a lower risk of cerebral complications

than symptomatic PAVM patients. Patients with symptomatic PAVM had more cerebral complications at the time of diagnosis (32.0% vs. 10.0%; P = 0.088). Additionally, three out of 15 untreated symptomatic PAVM patients developed cerebral complications, whereas one out of 18 untreated incidentally detected PAVM patients developed cerebral complications. The early detection of PAVM before complications develop may contribute to the low risk of cerebral complications in patients with incidentally detected PAVM.

The treatment of choice for patients with PAVMs is embolotherapy [1, 15, 21]. Surgical excision may be required when embolotherapy fails or pulmonary hemorrhage associated with PAVM rupture occurs [1, 15]. Embolotherapy improves oxygenation and exercise tolerance and decreases dyspnea, paradoxical embolization, strokes, migraines, and pulmonary hemorrhages [12]. Several studies have reported that PAVM treatment is associated with fewer cerebral complications [6, 9, 18, 26]. The untreated group had more cerebral complications than the treated group (P = 0.088) in this study. The treated group did not experience any cerebral complications during the follow-up period. In the older age group ($\geq$ 65 years), the odds of developing cerebral complications were 18 times higher than in the younger age group ($<$ 65 years) among patients who did not receive treatment after the PAVM diagnosis. Maher et al. also found that older patients with PAVM often experienced cerebral complications [14]. The high risk of cerebral complications in older patients with PAVMs should prompt them to seek active treatment.

This study had some limitations. First, this study was conducted retrospectively. Therefore, there is a limit to the generalizability of the findings. Second, there were very few patients with HHT compared with previous studies in this study. Because this is a retrospective study, HHT may have been underestimated. Third, a small number of 55 patients were enrolled in this 19-year retrospective study owing to the rarity of PAVM. Although we evaluated the functional disability associated with cerebral complications, the number of patients in this study was insufficient for statistical analysis. Therefore, prospective multicenter studies are required for a more robust evaluation. Fourth, diabetes and hypertension are well-known risk factors for stroke [27]. Hypertension and diabetes were not statistically significant in the univariate logistic regression analysis in this study. However, there were a few patients with hypertension or diabetes in this study cohort (11 with hypertension and three with diabetes). Fifth, the majority of strokes occur in people over 65. It is possible that observed cerebral complications cannot be definitively attributed to PAVM due to a small number of enrolled patients [28]. Therefore, multicenter studies that allow the inclusion of a larger number of patients with PAVMs are needed.

## Conclusions

The number of patients newly diagnosed with PAVM, particularly the number diagnosed incidentally without symptoms, is increasing. Older age ($\geq$65 years) was a risk factor for cerebral complications in patients with PAVMs; thus, treatment should be considered in older patients.

## Supporting information

**S1 File. Dataset.**
(XLSX)

**S1 Table. Characteristics of patients with cerebral complications.**
(DOCX)

## Author Contributions

**Conceptualization:** Young-Ok Na, Hong-Joon Shin, In-Jae Oh.

**Data curation:** Young-Ok Na, Hwa Kyung Park, Jae-Kyeong Lee, Bo-Gun Kho, Hyung-Joo Oh, Ha-Young Park.

**Formal analysis:** Bo-Gun Kho, Tae-Ok Kim, Yong-Soo Kwon, Yu-Il Kim, Sung-Chul Lim, Cheol-Kyu Park, In-Jae Oh, Young-Chul Kim, Ha-Young Park.

**Investigation:** Tae-Ok Kim, Hong-Joon Shin, Sung-Chul Lim.

**Methodology:** Yong-Soo Kwon, Yu-Il Kim, Sung-Chul Lim, Cheol-Kyu Park, In-Jae Oh, Young-Chul Kim.

**Project administration:** Young-Ok Na, Hwa Kyung Park, Jae-Kyeong Lee, Hyung-Joo Oh.

**Resources:** Young-Ok Na, Hong-Joon Shin.

**Supervision:** Hong-Joon Shin, Yong-Soo Kwon, Yu-Il Kim, Sung-Chul Lim, Cheol-Kyu Park, In-Jae Oh, Young-Chul Kim.

**Validation:** Ha-Young Park.

**Visualization:** Young-Ok Na, Hwa Kyung Park, Jae-Kyeong Lee, Bo-Gun Kho, Tae-Ok Kim, Hyung-Joo Oh.

**Writing – original draft:** Young-Ok Na, Hong-Joon Shin.

**Writing – review & editing:** Young-Ok Na, Hwa Kyung Park, Jae-Kyeong Lee, Bo-Gun Kho, Tae-Ok Kim, Hong-Joon Shin, Yong-Soo Kwon, Yu-Il Kim, Sung-Chul Lim, Hyung-Joo Oh, Cheol-Kyu Park, In-Jae Oh, Young-Chul Kim, Ha-Young Park.

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
