## [Decision Letter · Decision Letter 0]

3 May 2022

PONE-D-22-08058Risk factors for cerebral complications in patients with pulmonary arteriovenous malformations: a multicenter retrospective cohort studyPLOS ONE

Dear Dr. Shin,

Thank you for submitting your manuscript to PLOS ONE. After careful consideration, we feel that it has merit but does not fully meet PLOS ONE’s publication criteria as it currently stands. Therefore, we invite you to submit a revised version of the manuscript that addresses the points raised during the review process.

We look forward to receiving your revised manuscript.

Kind regards,

Alfred Pokmeng See, M.D.

Academic Editor

PLOS ONE

Journal Requirements:

[This study was supported by the National Research Foundation of Korea funded by the Korean Government (grant 2019R1F1A1060899).]

 [This study was supported by the National Research Foundation of Korea funded by the Korean Government (grant 2019R1F1A1060899).]

Additional Editor Comments:

Dear Dr. Shin,

Thank you for your submission regarding cerebral complications of pulmonary AVMs. Two reviewers from different subspecialty backgrounds have considered the manuscript and offer meaningful suggestions. A central concern is the specification of neurologic complications and use of functional outcome metrics, which would be standard in stroke reporting. Not all "cerebral complications" are equivalent and they may be unrelated to the pulmonary AVM. The reviewers also provide useful suggestions to conform with typical reporting in clinical neurology research.

Reviewers' comments:

Reviewer's Responses to Questions

**Comments to the Author**

1. Is the manuscript technically sound, and do the data support the conclusions?

Reviewer #1: Partly

Reviewer #2: Partly

2. Has the statistical analysis been performed appropriately and rigorously? 

Reviewer #1: Yes

Reviewer #2: Yes

3. Have the authors made all data underlying the findings in their manuscript fully available?

Reviewer #1: Yes

Reviewer #2: Yes

4. Is the manuscript presented in an intelligible fashion and written in standard English?

Reviewer #1: Yes

Reviewer #2: Yes

5. Review Comments to the Author

Reviewer #1: The authors seek to address an important issue in PAVM pathophysiology—the risk of developing cerebral complications—especially when the primary pathology is asymptomatic and incidental. The authors acknowledge their biggest limitation which is the small sample size of their population. Unfortunately, the small sample size underpowers their statistical analysis as can be seen in the wide range of their confidence interval. As such, I am concerned that their claims, especially with their subgroup analysis, is overstated. Nonetheless, this serves as a starting point for future studies. I would recommend that the authors strengthen their focus on the types of neurologic complications and highlight patient outcomes before this paper is suitable for acceptance.

Comments:

1. Study Design: I think the authors need to provide more details on how the patients were selected. Why were these patients with “incidental PAVMs” who are presumably asymptomatic scanned w a CT or pulmonary angiography? Would that introduce a population bias?

2. Results line 110-111: Since the authors want to describe the risk of cerebral complications w PAVMs, I think it is important for them to describe the types of cerebral complications they had in their populations. They only provide a binary description of symptomatic and incidentally detected.

3. Results line 115-116: Would recommend describing the complications that occurred in the patients who did not undergo treatment (were they considered more severe? Any pattern?)

4. Results line 115-116: The authors state a total of 13 pts had cerebral complications and of them, four patients did not undergo treatment. Subsequently, in line 117-118, they state that none of the patients who underwent treatment for PAVMs had cerebral complications. Please provide clarification or additional details on the remaining 9 patients with complications.

5. Would recommend providing a table detailing the demographics of the patients with cerebral complications and their outcomes

6. Table 2 reports several risk factors based on characteristics of PAVMs (feeding artery diameter, types of AVM) but does not report size or location which may be useful to include.

7. Figure 2: Is a Kaplan Meier survival curve the best choice in trying to highlight the morbidity of cerebral complications especially since there was no survival difference? It may be more meaningful to look at functional scales such as mRS or GOS.

8. Line 191: Missing the word echo for TTCE

Reviewer #2: In the conclusions section, there is no mention of the possibility that both hypertension and diabetes may also be a contributing factor given that one or other were present in all patients with cerebral complications with pAVM. Perhaps comment as to why it is not relevant or consider that in fact may well be.

6. PLOS authors have the option to publish the peer review history of their article (what does this mean?). If published, this will include your full peer review and any attached files.

Reviewer #1: No

Reviewer #2: No

---

## [Author Response · Author response to Decision Letter 0]

12 Jun 2022

PONE-D-22-08058

Risk factors for cerebral complications in patients with pulmonary arteriovenous malformations: a multicenter retrospective cohort study

PLOS ONE

Responses to reviewers

Reviewer #1: The authors seek to address an important issue in PAVM pathophysiology—the risk of developing cerebral complications—especially when the primary pathology is asymptomatic and incidental. The authors acknowledge their biggest limitation which is the small sample size of their population. Unfortunately, the small sample size underpowers their statistical analysis as can be seen in the wide range of their confidence interval. As such, I am concerned that their claims, especially with their subgroup analysis, is overstated. Nonetheless, this serves as a starting point for future studies. I would recommend that the authors strengthen their focus on the types of neurologic complications and highlight patient outcomes before this paper is suitable for acceptance.

A. Thank you for your precise and kind comments. This study only included a small number of participants because PAVM is a rare disease. We believe this study can serve as a starting point for diagnosing and treating of PAVM. We tried to strengthened the types and outcomes of neurologic complications, as you pointed out.

Q1. Study Design: I think the authors need to provide more details on how the patients were selected. Why were these patients with “incidental PAVMs” who are presumably asymptomatic scanned w a CT or pulmonary angiography? Would that introduce a population bias?

A1. Thank you for your comment. We screened patients with PAVM confirmed by diagnostic code, chest computed tomography (CT) findings, or pulmonary angiography findings during the study period. A total of 70 individuals were screened, of which 15 were excluded due to the following reasons: CT or angiography did not reveal PAVM (n = 11), and diagnosis of pulmonary sequestration (n = 3) or pulmonary venous anomaly (n = 1). A total of 55 patients with PAVM were enrolled in this study. 

We added this information to method and results sections. See line 57-59 on page 4 and line 109-112 on page 7.

Q2. Results line 110-111: Since the authors want to describe the risk of cerebral complications w PAVMs, I think it is important for them to describe the types of cerebral complications they had in their populations. They only provide a binary description of symptomatic and incidentally detected.

A2. Thank you for your valuable comment. We added the types of complications in the results section, as your recommendation. See line 120-123 on page 7.

Q3. Results line 115-116: Would recommend describing the complications that occurred in the patients who did not undergo treatment (were they considered more severe? Any pattern?)

A3. Thank you for your valuable comments. Cerebral complication in all the four patients included strokes, and one of them showed severe functional impairments, with a mRS score of 5. The other three, however, only showed mild impairments with mRS scores of 1¬3. We added this information to method and results sections. See line 126-132 on page 7-8.

Q4. Results line 115-116: The authors state a total of 13 pts had cerebral complications and of them, four patients did not undergo treatment. Subsequently, in line 117-118, they state that none of the patients who underwent treatment for PAVMs had cerebral complications. Please provide clarification or additional details on the remaining 9 patients with complications.

A4. Thank you for your valuable comments. In this study, of the 13 patients with cerebral complications, 12 had stroke and one had a cerebral abscess; moreover, six developed cerebral complications before PAVM diagnosis, and five developed them concurrently with PAVM diagnosis. Among patients with untreated PAVM, four patients developed cerebral complications after diagnosis of PAVM, and two of those had previous cerebral complications. We modified the results section as your recommendation. See line 120-123 on page 7.

Q5. Would recommend providing a table detailing the demographics of the patients with cerebral complications and their outcomes

A5. Thank you for your comments. We added a S1 Table for 13 patients with cerebral complications. See S1 Table. 

Q6. Table 2 reports several risk factors based on characteristics of PAVMs (feeding artery diameter, types of AVM) but does not report size or location which may be useful to include.

A6. Thank you for your valuable comments. PAVM locations and size of venous sac have been added to tables 1 and 2. However, both variables were not risk factors for development of cerebral complications in the univariate logistic regression analysis. See table 1 and 2 on page 8 and 11, respectively.

Q7. Figure 2: Is a Kaplan Meier survival curve the best choice in trying to highlight the morbidity of cerebral complications especially since there was no survival difference? It may be more meaningful to look at functional scales such as mRS or GOS.

A7. Thank you for your comments. Figure 2 shows a Kaplan-Meier curve of time without cerebral complications following treatment of pulmonary arteriovenous malformation. Although the untreated group developed more cerebral complications during the follow-up period, there was no significant difference between the two groups (log-rank P = 0.088). Due to the small number of patients enrolled, we did not consider the difference to be statistically significant. 

Based on your opinion, we measured the modified Rankin Scale (mRS) of patients with cerebral complications after detection of PAVM. In four patients, cerebral complications developed after PAVM detection, and the mRS varied from 1, 1, 3, and 5, respectively, therefore the number of patients was insufficient for a statistical analysis. These findings have been added as limitation part. See line 260-262 page on 17.

Q8. Line 191: Missing the word echo for TTCE

A8. I appreciate your kind words. The error has been corrected.

Reviewer #2 

Q1. In the conclusions section, there is no mention of the possibility that both hypertension and diabetes may also be a contributing factor given that one or other were present in all patients with cerebral complications with pAVM. Perhaps comment as to why it is not relevant or consider that in fact may well be.

A1. Thank you for your valuable comments. Diabetes and hypertension are well-known risk factors for stroke. In this study, we investigated whether diabetes and hypertension were risk factors for stroke among PAVM patients. Table 2 of this study shows risk factors associated with cerebral complications. Hypertension and diabetes were not statistically significant in univariate logistic regression analysis in this study. However, there were a few patients with hypertension or diabetes in this study cohort (11 with hypertension and three with diabetes). Therefore, multicenter studies that allow the inclusion of a larger number of patients with PAVMs are needed. This was added to the limitation. See line 263-267 on page 17.

---

## [Editor Report · Decision Letter 1]

28 Jun 2022

PONE-D-22-08058R1Risk factors for cerebral complications in patients with pulmonary arteriovenous malformations: a multicenter retrospective cohort studyPLOS ONE

Dear Dr. Shin,

Thank you for submitting your manuscript to PLOS ONE. After careful consideration, we feel that it has merit but does not fully meet PLOS ONE’s publication criteria as it currently stands. Therefore, we invite you to submit a revised version of the manuscript that addresses the points raised during the review process.

We look forward to receiving your revised manuscript.

Kind regards,

Alfred Pokmeng See, M.D.

Academic Editor

PLOS ONE

Journal Requirements:

Additional Editor Comments:

Before requesting further input from the reviewers, please further address the following questions:

Reviewer 1

Q1 - The reviewer asks the medical rationale for workup and diagnosis of patients without symptoms that led to PAVM. In lines 57-59, you provide information on how these patients were identified from the medical record system, but not why they were evaluated with CT or angiography. "More than half of the enrolled patients were diagnosed with PAVMs incidentally without symptoms."

Q2. - The reviewer asks for more specifics of the type of cerebral complication, and you now explain stroke vs. abscess. More detail is likely appropriate again, from the perspective of this reviewer as a neuroscience expert. For example, presumably these are ischemic strokes, although hemorrhagic stroke may also be possible. Furthermore, in discussing cerebral complications at the time of diagnosis (13 patients), the authors now report 9 symptomatic vs 4 asymptomatic, and 12 strokes vs 1 abscess, but describe 6 before diagnosis and 5 at the time of diagnosis (adding up to 11, not 13). The characterization of complications before pAVM diagnosis merits further clarification in the methods. Are these prior strokes based on brain imaging (not described in methods)?

Furthermore, table 1 shows 4 patients with neurologic symptoms. How do these relate to the 9symptomatic/4asymptomatic or 6before/5concurrent proportions described in the prose?

Q2-4. - The reviewer asks for a more direct description of:

pAVM - presenting asymptomatically

pAVM - presenting with symptoms unrelated to the brain

pAVM - presenting with neurologic symptoms

pAVM untreated patients - risk of subsequent neurologic symptoms

pAVM treated patients - risk of subsequent neurologic symptoms

Part of the problem appears to be semantic. The authors use the same term, cerebral complication, to describe strokes before diagnosis, strokes which were the presenting symptom leading to diagnosis, and strokes which occur after diagnosis. These are then used in one large paragraph. A flow chart of the events or a timeline plot of events may provide better clarity. For example Figure 1 in https://www.ncbi.nlm.nih.gov/pmc/articles/PMC7606530/

Q5. - S1 provides no legend for interpretation of the coded data or column names. For example, column D "Detect_reasons" has codes 1-9. The reader would benefit from seeing the 9 reasons that pAVM is presenting for evaluation. Table 1 only has 8 'causes of detection'

Reviewer 1 had some concerns about internal consistency of the data in the form presented. Improved clarity of the terminology and better specification of the timeline will likely benefit the manuscript before return to reviewers.

---

## [Author Response · Author response to Decision Letter 1]

22 Aug 2022

Responses to reviewers

Reviewer #1: The authors seek to address an important issue in PAVM pathophysiology—the risk of developing cerebral complications—especially when the primary pathology is asymptomatic and incidental. The authors acknowledge their biggest limitation which is the small sample size of their population. Unfortunately, the small sample size underpowers their statistical analysis as can be seen in the wide range of their confidence interval. As such, I am concerned that their claims, especially with their subgroup analysis, is overstated. Nonetheless, this serves as a starting point for future studies. I would recommend that the authors strengthen their focus on the types of neurologic complications and highlight patient outcomes before this paper is suitable for acceptance.

A. Thank you for your precise and kind comments. This study only included a small number of participants because PAVM is a rare disease. We believe this study can serve as a starting point for diagnosing and treating of PAVM. We tried to strengthened the types and outcomes of neurologic complications, as you pointed out.

Q1. Study Design: I think the authors need to provide more details on how the patients were selected. Why were these patients with “incidental PAVMs” who are presumably asymptomatic scanned w a CT or pulmonary angiography? Would that introduce a population bias?

A1. Thank you for your comment. 

We screened patients with PAVMs using diagnostic code, chest computed tomography (CT) findings, or pulmonary angiography findings during the study period. A total of 70 individuals were screened, of which 15 were excluded for the following reasons: a CT or angiography did not reveal PAVM (n = 11), or they were diagnosed with pulmonary sequestration (n = 3) or a pulmonary venous anomaly (n = 1). Therefore, a total of 55 patients with PAVM were enrolled in this study. [See line 57-59 on page 4 and line 118-121 on page 7]

A definition of incidentally detected PAVM has been added to the method section as "Patients with incidentally detected PAVM were those whose PAVM was detected during a health screening, preoperative evaluation, or evaluation of other diseases, and presented asymptomatically. ". [See line 84-57 on page 5] 

Thirty patients were incidentally detected PAVM due to health screening (n = 18), preoperative evaluation (n = 6), and evaluation of other diseases (n = 6) such as hepatocellular carcinoma, Sjogren syndrome, External MALToma, AOSD, back pain, or hoarseness. These findings have also been added to the results section [See line 127-129 on page 7]

Q2. Results line 110-111: Since the authors want to describe the risk of cerebral complications w PAVMs, I think it is important for them to describe the types of cerebral complications they had in their populations. They only provide a binary description of symptomatic and incidentally detected.

A2. Thank you for your valuable comment. There were 11 patients with cerebral ischemia and one patient with cerebral hemorrhage. We change the manuscript as “A total of thirteen patients (23.6%) had cerebral complications, including cerebral ischemia (n = 11), cerebral hemorrhage (n = 1), and cerebral abscess (n = 1).”. [See line 129-130 on page 7] We added the types of complications in the results section, as your recommendation. See line 120-123 on page 7.

Q3. Results line 115-116: Would recommend describing the complications that occurred in the patients who did not undergo treatment (were they considered more severe? Any pattern?)

A3. Thank you for your valuable comments. Cerebral complications occurred in four patients who did not undergo treatment after a PAVM diagnosis, and two patients had previous cerebral complications. Cerebral complications in all four patients included cerebral ischemia, and one patient showed severe functional impairments, with an mRS score of 5. The other three patients, however, only showed mild impairments with mRS scores of 1¬3 (S1 Table). We added this information to method and results sections. [See line 136-141 on page 7-8.

Q4. Results line 115-116: The authors state a total of 13 pts had cerebral complications and of them, four patients did not undergo treatment. Subsequently, in line 117-118, they state that none of the patients who underwent treatment for PAVMs had cerebral complications. Please provide clarification or additional details on the remaining 9 patients with complications.

A4. Thank you for your valuable comments. 

A total of thirteen patients (23.6%) had cerebral complications, including cerebral ischemia (n = 11), cerebral hemorrhage (n = 1), and cerebral abscess (n = 1). Of the 13 patients with cerebral complications, six developed these complications before their PAVM diagnosis, five developed these complications concurrently with their PAVM diagnosis, and two developed these complications after their PAVM diagnosis. 

 Cerebral complications occurred in four patients who did not undergo treatment after a PAVM diagnosis, and two patients had previous cerebral complications. 

We modified the results section as your recommendation. [See line 129-133 and 136-138 on page 7]

Q5. Would recommend providing a table detailing the demographics of the patients with cerebral complications and their outcomes

A5. Thank you for your comments. We added a S1 Table for 13 patients with cerebral complications. [See S1 Table] 

Q6. Table 2 reports several risk factors based on characteristics of PAVMs (feeding artery diameter, types of AVM) but does not report size or location which may be useful to include.

A6. Thank you for your valuable comments. PAVM locations and size of venous sac have been added to tables 1 and 2. However, both variables were not risk factors for development of cerebral complications in the univariate logistic regression analysis. [See table 1 and 2 on page 8 and 11, respectively]

Q7. Figure 2: Is a Kaplan Meier survival curve the best choice in trying to highlight the morbidity of cerebral complications especially since there was no survival difference? It may be more meaningful to look at functional scales such as mRS or GOS.

A7. Thank you for your comments. 

Figure 2 shows a Kaplan-Meier curve of time without cerebral complications following treatment of pulmonary arteriovenous malformation. Although the untreated group developed more cerebral complications during the follow-up period, there was no significant difference between the two groups (log-rank P = 0.088). Due to the small number of patients enrolled, we did not consider the difference to be statistically significant. 

Based on your opinion, we measured the modified Rankin Scale (mRS) of patients with cerebral complications after detection of PAVM. In four patients, cerebral complications developed after PAVM detection, and the mRS varied from 1, 1, 3, and 5, respectively, therefore the number of patients was insufficient for a statistical analysis. These findings have been added as limitation part. [See line 272-276 page on 18].

Q8. Line 191: Missing the word echo for TTCE

A8. I appreciate your kind words. The error has been corrected.[See line 218 on page 15]

Reviewer #2 

Q1. In the conclusions section, there is no mention of the possibility that both hypertension and diabetes may also be a contributing factor given that one or other were present in all patients with cerebral complications with pAVM. Perhaps comment as to why it is not relevant or consider that in fact may well be.

A1. Thank you for your valuable comments. Diabetes and hypertension are well-known risk factors for stroke. In this study, we investigated whether diabetes and hypertension were risk factors for stroke among PAVM patients. Table 2 of this study shows risk factors associated with cerebral complications. Hypertension and diabetes were not statistically significant in univariate logistic regression analysis in this study. However, there were a few patients with hypertension or diabetes in this study cohort (11 with hypertension and three with diabetes). Therefore, multicenter studies that allow the inclusion of a larger number of patients with PAVMs are needed. This was added to the limitation. See line 276-280 on page 18.

Responses to editor

Q1 - The reviewer asks the medical rationale for workup and diagnosis of patients without symptoms that led to PAVM. In lines 57-59, you provide information on how these patients were identified from the medical record system, but not why they were evaluated with CT or angiography. "More than half of the enrolled patients were diagnosed with PAVMs incidentally without symptoms."

A1. Thank you for your precise and kind comments. 

A definition of incidentally detected PAVM has been added to the method section as "Patients with incidentally detected PAVM were those whose PAVM was detected during a health screening, preoperative evaluation, or evaluation of other diseases, and presented asymptomatically. ". [See line 84-57 on page 5] 

Thirty patients were incidentally detected PAVM due to health screening (n = 18), preoperative evaluation (n = 6), and evaluation of other diseases (n = 6) such as hepatocellular carcinoma, Sjogren syndrome, External MALToma, AOSD, back pain, or hoarseness. These findings have also been added to the results section [See line 127-129 on page 7]

Q2-1- The reviewer asks for more specifics of the type of cerebral complication, and you now explain stroke vs. abscess. More detail is likely appropriate again, from the perspective of this reviewer as a neuroscience expert. For example, presumably these are ischemic strokes, although hemorrhagic stroke may also be possible. 

A2-1. Thank you for your comments. There were 11 patients with cerebral ischemia and one patient with cerebral hemorrhage. We change the manuscript as “A total of thirteen patients (23.6%) had cerebral complications, including cerebral ischemia (n = 11), cerebral hemorrhage (n = 1), and cerebral abscess (n = 1).”. [See line 129-130 on page 7]

Q2-2. Furthermore, in discussing cerebral complications at the time of diagnosis (13 patients), the authors now report 9 symptomatic vs 4 asymptomatic, and 12 strokes vs 1 abscess, but describe 6 before diagnosis and 5 at the time of diagnosis (adding up to 11, not 13). 

A2-2. Thank you for your comments. 

It has caused confusion to use words such as "incidentally detected", "symptomatic", and "asymptomatic.". The term "asymptomatic patients" meant the same thing as incidentally detected patients. 

We added the definition of incidentally detected PAVM on the method section. [See page 5, line 84-87] In addition, sentences with unclear meanings were removed. [See page 7, line 131-133], and see page 8, line 144-145 of revised manuscript with track change file]

The discussion section was also edited. [See line 213-217 on page 15]

Q2-3. The characterization of complications before pAVM diagnosis merits further clarification in the methods. 

A2-3. Thank you for your comment. We added the clarification of cerebral complications to the method section as “The cerebral complications associated with PAVM were divided into cerebral ischemia, hemorrhage, and abscess.”. [See line 74-75 on page 4]

Q2-4. Are these prior strokes based on brain imaging (not described in methods)?

A2-4. Thank you for your comment. In order to determine whether there had been any brain complications before PAVM diagnosis, we reviewed the patient's medical records. However, brain CT or MRI was used to diagnose cerebral complications associated with PAVM at the time or after detected PAVM. We modified the method section as “A brain CT or MRI was used to diagnose cerebral complications associated with PAVM at the time of initial PAVM diagnosis or after PAVM was detected.” [See line 76-78 on page 5]

Q2-5. The reviewer asks for a more direct description of:

pAVM - presenting asymptomatically

pAVM - presenting with symptoms unrelated to the brain

pAVM - presenting with neurologic symptoms

pAVM untreated patients - risk of subsequent neurologic symptoms

pAVM treated patients - risk of subsequent neurologic symptoms

Part of the problem appears to be semantic. The authors use the same term, cerebral complication, to describe strokes before diagnosis, strokes which were the presenting symptom leading to diagnosis, and strokes which occur after diagnosis. These are then used in one large paragraph. A flow chart of the events or a timeline plot of events may provide better clarity. For example Figure 1 in https://www.ncbi.nlm.nih.gov/pmc/articles/PMC7606530/

A2-5. Thank you for your valuable comments. 

There was some confusion in the terminology we used, as you pointed out. In the method section, "PAVM presenting asymptomatically" is defined as “Patients with incidentally detected PAVM were those whose PAVM was detected during a health screening, preoperative evaluation, or evaluation of other diseases, and presented asymptomatically.”. [See line 85-57 on page 5]

 In our study, we did not distinguish between PAVMs with symptoms unrelated to the brain and PAVMs with neurologic symptoms. 

We clarified the cerebral complications associated with PAVM. We evaluated the risk factors for cerebral complications, including cerebral hemorrhage and ischemia, at the time of PAVM diagnosis (Table 2). Figure 2 shows the Kaplan–Meier curve for the development of new cerebral complications in the treated and untreated groups after PAVM diagnosis. We investigated the risk factors associated with new cerebral complications in 33 patients who did not receive treatment after their PAVM diagnosis (Table 3). Figure 3 shows the Kaplan–Meier curve for the development of new cerebral complications in the ≥ 65 years and < 65 years age groups of patients who did not undergo treatment after a PAVM diagnosis. The results section has been edited. [See line 159-160 on page 10, line 171-172 on page 12, line 177-183 on page 12, line 187-188 on page 13, line 191-193 on page 13, line 199-200 on page 14]

Despite your recommendation, we were not able to create a timeline plot because the timing of past cerebral complications was unclear (retrospective study).

Q3. - S1 provides no legend for interpretation of the coded data or column names. For example, column D "Detect_reasons" has codes 1-9. The reader would benefit from seeing the 9 reasons that pAVM is presenting for evaluation. Table 1 only has 8 'causes of detection'

A3. Thank you for your comments. We modified detect reasons of S1 files. The reason 1 and 2 are belongs to incidentally categories (1: health screening, 2: preoperative evaluation or other disease evaluations).

---

## [Decision Letter · Decision Letter 2]

24 Oct 2022

PONE-D-22-08058R2Risk factors for cerebral complications in patients with pulmonary arteriovenous malformations: a multicenter retrospective cohort studyPLOS ONE

Dear Dr. Shin,

Thank you for submitting your manuscript to PLOS ONE. After careful consideration, we feel that it has merit but does not fully meet PLOS ONE’s publication criteria as it currently stands. Therefore, we invite you to submit a revised version of the manuscript that addresses the points raised during the review process.

We look forward to receiving your revised manuscript.

Kind regards,

Alfred Pokmeng See, M.D.

Academic Editor

PLOS ONE

Journal Requirements:

Additional Editor Comments:

Thank you for your revisions. Due to reviewer availability, additional reviewers were requested. They shared a similar concern regarding the possibility of comorbid risks for stroke in the older patient population. It would be appropriate to explicitly state in the limitations that the observed cerebral complications cannot be definitively linked to the PAVM, although this is assumed in the analysis.

For Table 3 the multivariate analysis section and the related results text only applies the statistical test (multivariate logistic regression) to a single variable (Age) therefore, it is not different from the univariate logistic regression, and logically, reports the same OR, CI, and p-value. Since no other co-variates were included in the 'multivariate' analysis, it is misleading to present this as a multivariate analysis and to re-present the data in a separate section of the table.

Reviewers' comments:

Reviewer's Responses to Questions

**Comments to the Author**

1. If the authors have adequately addressed your comments raised in a previous round of review and you feel that this manuscript is now acceptable for publication, you may indicate that here to bypass the “Comments to the Author” section, enter your conflict of interest statement in the “Confidential to Editor” section, and submit your "Accept" recommendation.

Reviewer #3: All comments have been addressed

2. Is the manuscript technically sound, and do the data support the conclusions?

Reviewer #3: No

3. Has the statistical analysis been performed appropriately and rigorously? 

Reviewer #3: No

4. Have the authors made all data underlying the findings in their manuscript fully available?

Reviewer #3: No

5. Is the manuscript presented in an intelligible fashion and written in standard English?

Reviewer #3: Yes

6. Review Comments to the Author

Reviewer #3: Cerebral and hemorrhagic complications associated with PAVMs are often fatal. This study retrospectively evaluated the risk factors for cerebral complications in patients with PAVMs. The included cerebral complications complications were stroke (12 cases) and brain abscess (one case ). The study was retrospective, and it is doubtful whether stroke is caused by PAVM in elderly patients.

7. PLOS authors have the option to publish the peer review history of their article (what does this mean?). If published, this will include your full peer review and any attached files.

Reviewer #3: No

---

## [Author Response · Author response to Decision Letter 2]

15 Nov 2022

<Response to editor>

Q1) Please review your reference list to ensure that it is complete and correct. If you have cited papers that have been retracted, please include the rationale for doing so in the manuscript text, or remove these references and replace them with relevant current references. Any changes to the reference list should be mentioned in the rebuttal letter that accompanies your revised manuscript. If you need to cite a retracted article, indicate the article’s retracted status in the References list and also include a citation and full reference for the retraction notice.

A1) Thank you for your comments. All references cited in this article have been thoroughly reviewed. We have also mentioned additional references below.

Additional Editor Comments:

Q2) Thank you for your revisions. Due to reviewer availability, additional reviewers were requested. They shared a similar concern regarding the possibility of comorbid risks for stroke in the older patient population. It would be appropriate to explicitly state in the limitations that the observed cerebral complications cannot be definitively linked to the PAVM, although this is assumed in the analysis.

A2) Thank you for your comments. We understand the concerns of reviewers and editors. We have added these concerns to the limitation section with reference as your suggestion. (See line 278-280 on page 18.)

Q3) For Table 3 the multivariate analysis section and the related results text only applies the statistical test (multivariate logistic regression) to a single variable (Age) therefore, it is not different from the univariate logistic regression, and logically, reports the same OR, CI, and p-value. Since no other co-variates were included in the 'multivariate' analysis, it is misleading to present this as a multivariate analysis and to re-present the data in a separate section of the table.

A3) Thank you for your valuable comments. We mentioned follows in the statistical analysis part of the methods section: Subsequent multivariate logistic regression analyses included variables with P values < 0.1 in the univariate analysis using a backward method. We changed the variable to be selected in univariate analysis from P <0.1 to P <0.2 in the same way as in Cox-regression analysis. As a result, age >65 and hypertension were selected in the univariate analysis. Table 3 shows the results of multivariate analysis with age >65 and hypertension. Additionally, diabetes was also selected in univariate analysis in Table 2. After multivariate analysis, the results are presented in table 2. (See line 34 on page 2, 110 on page 6, line 162-164 on page 10-11, table 2, line 185 on page 13, and table 3)

* Reference 21 has been added as a recent article. (See line 217-219 on page 15, and line 255 on page 17) 

<Response to Reviewer #3>

Q1) Reviewer #3: Cerebral and hemorrhagic complications associated with PAVMs are often fatal. This study retrospectively evaluated the risk factors for cerebral complications in patients with PAVMs. The included cerebral complications complications were stroke (12 cases) and brain abscess (one case ). The study was retrospective, and it is doubtful whether stroke is caused by PAVM in elderly patients.

A1) Thank you for your valuable comments. In this study, we understand the concerns of older patients. The risk of stroke in elderly patients is well known. Further studies are needed to determine whether strokes are more common in elderly patients with pulmoanry AVMs. We have added these concerns to the limitation section. (See line 278-280 on page 18.)

---

## [Editor Report · Decision Letter 3]

21 Nov 2022

Risk factors for cerebral complications in patients with pulmonary arteriovenous malformations: a multicenter retrospective cohort study

PONE-D-22-08058R3

Dear Dr. Shin,

We’re pleased to inform you that your manuscript has been judged scientifically suitable for publication and will be formally accepted for publication once it meets all outstanding technical requirements.

Kind regards,

Alfred Pokmeng See, M.D.

Academic Editor

PLOS ONE

Additional Editor Comments (optional):

Please correct the grammatical error in the revision:

Fifth, majority of strokes occur in people over 65.

Fifth, the majority of strokes occur in people over 65.
---

## [Editor Report · Acceptance letter]

23 Nov 2022

PONE-D-22-08058R3 

Risk factors for cerebral complications in patients with pulmonary arteriovenous malformations: a multicenter retrospective cohort study 

Dear Dr. Shin:

I'm pleased to inform you that your manuscript has been deemed suitable for publication in PLOS ONE. Congratulations! Your manuscript is now with our production department. 

Kind regards, 

on behalf of

Dr. Alfred Pokmeng See 

Academic Editor

PLOS ONE